

# Earthquake Response Timeliness: Disaster Managers Experience in Responding to Earthquakes in Iran

**Reza Hassanzadeh**
Department of Ecology, Institute of Science and High Technology and Environmental Sciences, Graduate University of
Advanced Technology, Kerman-Iran.
Corresponding author: Reza Hassanzadeh, Email: Hassanzadeh22@yahoo.com

**Abstract.** This paper explores the timeliness of main response activities in the Bam earthquake in order to shed light in disaster response effectiveness in the aftermath of the earthquake. Disaster managers who were involved in disaster response activities in the Bam earthquake were selected using snowball sampling method and 30 participants were interviewed. The interviews data were analyzed applying thematic analysis method. The results showed that the extensive damage and large number of casualties challenged the local emergency response services. The problem was compounded by the lack of personnel in the first few hours after the earthquake. Iran's disaster management system was not prepared to cope with a disaster of this scale. According to the interview data, a response timeline to the earthquake was developed for 48 hours after the earthquake that can assist disaster manager in designing a proper action plan (strategic response plan) prior to an earthquake in the region. This is an efficient way of dealing with disaster response challenges in the aftermath of earthquakes.

**Keywords:** Response timeliness, Semi-structured interviews, Thematic method, Disaster managers, and the Bam earthquake.
**1. Introduction: Timeliness as a factor in measuring disaster response effactivness**
This paper explores the timeliness of main activities that have been operated in response to the Bam earthquake in order to shed
light in disaster response effectiveness in the aftermath of the earthquake. The focus is primarily on timeliness in relation to the
first 48 hours of response and more on national and local responders.
The disaster management cycle is a complex process that should be addressed clearly to facilitate all phases of disaster management
including prevention and mitigation, preparation (preparedness), response, and recovery (Alexander, 1993; Clary, 1985; Cutter,
2003).The disaster-response phase focuses on conducting operations to assist the affected community during a disaster and in its
aftermath (Alexander, 1993; Clary, 1985; Cutter, 2003). In this stage, responders provide emergency assistance to victims and act
to reduce the likelihood of secondary damage in the area. In the first few hours of a disaster, a proper assessment of the situation
is particularly important. Many questions need to be answered: Where is the emergency? How can responders get to the damaged
area? How many casualties are there? (Diehl et al., 2006; Quarantelli, 1997). However, the critical component of any disaster
response is an early and accurate assessment to detect damage, to identify urgent needs, and to determine relief priorities for the
affected population (Lillibridge et al., 1993; Gunawan et al., 2011)  wich they need accuracte and high quality information for





decision making during disaster respons (Jayawardene et al., 2021). Abir et al. (2017) developed 5 main factors in measuring
disaster response performance in the aftermath of a disaster. These include "(1) timeliness and efficiency, (2) supplies and
equipment, (3) transportation, (4) personnel, and (5) interagency cooperation" (Abir et al., 2017). Moreover, effective decision-
making and timely response operations are important in facilitating rapid responses to a disaster (Borkulo et al., 2006; Ouyang et
al., 2008; Smilowitz and Dolinskaya, 2011). For example Zhang et al. (2002) found that accessible, up-to-date, accurate data, and
speedy data analysis by disaster managers could facilitate the decision-making process and lead to rapid responses to emergencies
(Leidner et al., 2009; Mansourian et al., 2005). Another major factor in the response phase is collaboration among the emergency
services involved in disaster response (Scotta et al., 2007). As seen in previous disasters such as the 1995 Kobe earthquake, rapid
fire-fighting operations help to save lives and protect property (Namba et al., 1997). In addition, police presence play an important
role in controlling and monitoring public security (Varano et al., 2010; Sullum et al., 2010). Another aspect that can lead to an
effective disaster response is community involvement in responding to a disaster (Colina et al., 2004; Zhang et al., 2013). Therefore,
response operations can be challenging due to lack of information, absence of required resources, and limited access to damaged
areas, violation of security and disruption of communication systems, outbreaks of fire, and massive numbers of fatalities (Auf
Der Heide, 1989; Chen et al., 2011; National Disater Managemrnt Organization (Ndmo), 2011; Nekoei-Moghadam et al., 2016).
All of the challenges described above show the complexity of the response phase.
This research aim is to measure timeliness in disaster response activities, as seen in figure 1. A response timeline that is created
according to the activities that have been performed in the aftermath of an earthquake can assist disaster manager in evaluating
their response to the earthquake. For example, a response timeline that was developed for Hyogo prefecture in Japan based on the
records of the 1995 Great Hanshin-Awaji Eerthquake, the 2011 Great East Japan earthquake, and the Awaji Island earthquake in
2013 that it provided useful idea for disaster manager regarding what to do and how to do it after a disaster (Kikakuka, 2015). In
another example, a response timeline was presented regarding government's response in the aftermath of the Nepal (Gorkha)
earthquakes in 2015 (Gsma, 2015; United Nations Development Programme (Undp), 2016; Earthquakes and Megacities Initiative
(Emi), 2015) and also a timeline that was reported on the parliament's response to the Canterbury earthquake in New Zealand
(New Zealand Parliament, 2010). These response timelines can help disaster managers in designing or revising the strategic
response plans in case of any future earthquakes occurrence. However, in case of the Bam earthquake, there was lack of studies on
producing a response timeline for this earthquake, and this is the motive behind conducting this research. Therefore, this study
investigates the timeliness of response activities to the affected area by conducting semi-structured interviews with disaster
managers who were involved in the Bam earthquake.

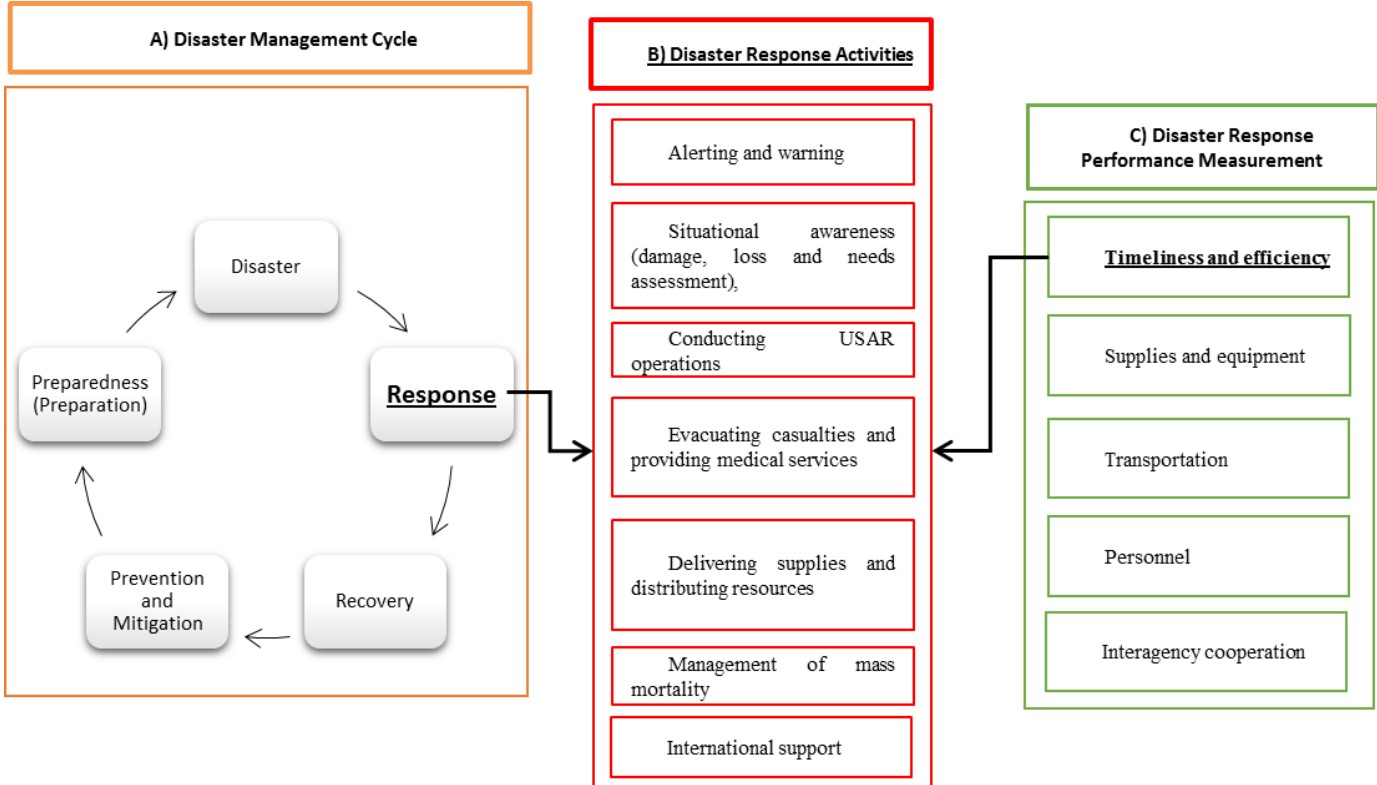

Figure 1.A. Disaster management cycle (Alexander, 1993), B. Disaster response activities and C) Performance measurement
(Abir et al., 2017)

## 2. Material and methods

This study was based on a qualitative research approach, and data were collected through reviewing literature and interviewing
disaster managers to clarify the timeliness of main activities in disaster response. This study has been conducted through several
stages as explained in the following: Selecting the Bam earthquake as a case study, collecting data by interviewing disaster
responders including managers and experts who were involved in disaster response activities during the Bam earthquake,
conducting thematic analysis method, and interpreting the results by considering other studies.

### 2.1. Case study

Bam city that was struck by a devastating earthquake in 2003 (Ahmadizadeh and Shakib, 2004) has been selected as a case
study. It is located in the southeast of Iran (Figure 2). This city was selected because of the existence of an adequate data and
literatures regarding the impact of earthquake on buildings, population and infrastructure, and availability of disaster managers
who were involved in disaster-response in the earthquake for conducting interview surveys.

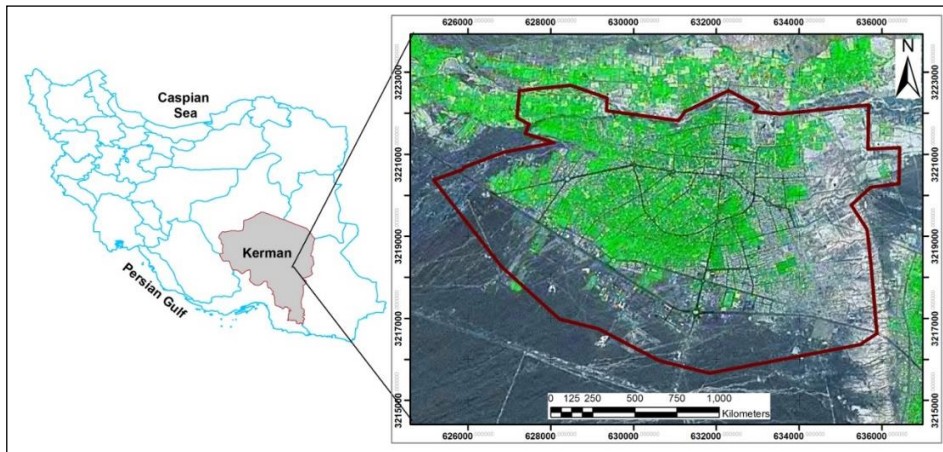


Figure 2.Geographical location of Bam city

The Bam earthquake occurred on 26 December 2003, at 05:56:56 local time (01:56:56 Greenwich mean-time) when many
people were sleeping in their homes. The magnitude was reported at 6.3 mb (body wave magnitude) by the Geophysical Institute
of Tehran University (Geophysical Institute of Tehran University (Gitu), 2003). Surface wave magnitude measured 6.7 Ms and
moment magnitude was determined as 6.6 Mw by the US Geological Survey (U.S. Geological Survey (Usgs), 2003; Ahmadizadeh
and Shakib, 2004). Moreover, two foreshocks occurred the night before the earthquake, at approximately 15:00 and 22:00 local
time, and another foreshock was felt 35 minutes before the main shock, with a magnitude of less than 4 Richter (Berberian, 2009).
Zare (2004) mentioned 5 foreshocks greater than 5 Mb just seconds before the main earthquake.
The earthquake inflicted considerable impact on the population and buildings. The infrastructures and also historical places
were badly affected (Ghafory-Ashtiany and Hosseini, 2008).  The Statistical Centre of Iran (Statistical Centre of Iran (Sci), 2004)
collected the post-earthquake data on the impacts of the earthquake on the population and buildings of Bam city via a street survey
in 2004. According to this information, 19,087 families (89,145 people), were living in Bam city at the time of the earthquake. As
a result of the earthquake, 22,391 people were killed, 8,136 were injured and hospitalized, 55,167 were not injured, and 422 people
were missing (Statistical Centre of Iran (Sci), 2004).
The total number of buildings in the affected region was 45,395, of which 34,093 collapsed, while the remaining 10,827 were
partially damaged. These included 131 school and healthcare buildings in the Bam region (Statistical Centre of Iran (Sci), 2004).
In Bam city, there were a total of 28,625 buildings and gardens in Bam city, of which 26,111 collapsed and 2,381 were partially
damaged (Statistical Centre of Iran (Sci), 2004). From the total number of buildings and gardens, 22,585 units were individual
buildings and the remaining were included in the gardens (The level of destruction of a garden mostly related to its surrounding
walls which can cause road blockages). According to the building damage map produced by the National Cartographic Center of





Iran (National Cartographic Center (Ncc), 2003) via image processing of the aerial photos (taken 3 days after the earthquake),
about 80% of all buildings were destroyed (Earthquake Engineering Research Institute (Eeri), 2004).

## 2.2. Disaster management system in Iran

Management of natural disasters in Iran was under the responsibility of a special disaster task force in the Office of the President
before 1991. Since 1991, several changes were made in the disaster management structure, especially after the Manjil Earthquake
in 1990. On a national level, disaster risk management came under the Ministry of Interior (MOI). The responsibilities and
functions related to natural and human-made disasters were formally assigned to this Ministry by virtue of the Budget Act in 1991
(National Disater Managemrnt Organization (Ndmo), 2011). Two specialized bodies were created to provide support and to
operationalize the disaster management activities: A) the Bureau for Research and Coordination of Safety and Reconstruction
Affairs (BRCRS), which was tasked with mandates including research; formulation of preparedness and mitigation plans;
collection, analysis and dissemination of related information; coordination of relief, reconstruction and rehabilitation activities. B)
The National Disaster Task Force (NDTF), which was established as an inter-organizational coordinating body, particularly for
emergency relief operations in the national territory. Emergency response across sectors was under the responsibility of the
ministries that were coordinated by the NDTF. The manager of the NDTF was also the director of BRCRS. Based on the enactments
of the National Committee for Reduction of Natural Disaster act in 1991, the role of the Ministry of Interior and this committee
was established as a policy and decision-making body, with the mandate to research and explore practical ways to mitigate natural
disasters (National Disater Managemrnt Organization (Ndmo), 2011).
In 2003, the Rescue and Relief Comprehensive Plan (as an Integrated National Disaster Management Plan (INDMP)) was
approved by the Council of Minister's Decree. According to this plan, three major components were introduced: 1) A National
Disaster Task Force (NDTF), chaired by the Minister of the Interior. The main functions of the NDTF were policy-making,
decision-making and coordination of different organizations' activities in the mitigation, preparedness, response, and recovery
phases of disaster management. 2) The National Committee for Preparedness against Natural Disaster (NCPND) was responsible
for training, research, planning, and exercises for disaster management. There were three specialized working groups under this
committee, including operation, education, and prevention and management of disasters. 3) Specialized affairs that were
responsible for the other issues related to disaster management of the natural and human-made disasters. These components were
suggested at national, provincial, and township levels (National Disater Managemrnt Organization (Ndmo), 2011).
Since the Bam earthquake disaster in 2003, the Headquarter Council of Disaster Management and Prevention was established
by the Council of Minister's Decree in 2004 under the Presidential office. This decision allowed for effective and comprehensive
disaster management and coordination among all the organizations involved in disaster management (National Disater Managemrnt
Organization (Ndmo), 2011).



In 2008, under a new National Disaster Management Law passed by the Parliament, the National Disaster Management Organization (NDMO) was formed. NDMO aimed to utilize the national, regional and local capacities to cope with all forms of disasters. Moreover, it was to create an integrated management system for planning and coordinating executive activities in a cohesive manner over different phases of disaster management in affected areas (as it was discussed in the Rescue and Relief Comprehensive Plan in 2003). The objectives of NDMO were to utilize the full potential and resources available from government ministries, public institutions, and private sectors. These involved banks, insurance companies, military forces, non–governmental institutes, Islamic councils, municipalities, public associations, and organizations under the auspices of the Supreme Leader of the country. In order to coordinate the activities of the involved organizations (organization which were primarily responsible to manage a disaster) and institutions affiliated to the legislative, executive and judiciary organs, and the Armed Forces, and enacting of regulations and standards governing the four phases of disaster management (prevention and mitigation, preparedness, response and recovery), the Supreme Council of the National Disaster Management Organization was formed.

The Disaster Management Coordination Council has also been formed under the chairmanship of the Head of National Disaster Management Organization. Membership of the representatives deputizing the related organizations and agencies coordinated the activities regarding the four phases of disaster management. Disaster Management Coordination Council at provincial and city levels were formed under the chairmanship of the Governor Generals and Governors, respectively. The membership included all relevant organizations involved and twenty-two specialized groups from relevant ministries and organizations to cooperate with the Disaster Management Organization (Seyedin, 2005) (National Disater Managemrnt Organization (Ndmo), 2011) (Ali Ahamadi, 2012; Ardalan et al., 2012 ). Therefore, an integrated structure for disaster management was formed in 2008 in different administration levels, including national, provincial, township, and city levels (National Disater Managemrnt Organization (Ndmo), 2011) and the aw was passed in the parliament in 2019 (Parliment, 2019). The main changes were at the city level, where many organizations involved in disaster management should be prepared to cope with a probable disaster. All this happened as a results of disaster management challenges in responding to the Bam earthquake.

**2.3. Study Design**

Qualitative research methods were used to explore perceptions of disaster response timeliness among a group of disaster managers (people who made the decision on how to react) and experts that were involved in responding to the Bam earthquake in 2003 in order to generate more detailed insights into timeliness of the numerous response activities in the aftermath of the earthquake (the focus was primarily on Iranian actors). In doing so, in this research in-depth interview approach was selected. Although, focus groups can allow for the maximisation of data collection, particularly if time is limited (Acocella, 2012), but participants can hide the truth in presence of other colleagues (Sim, 1998). However, interviews can provide in-depth information and insight that cannot be discussed in a group of people coming from different organizations (Sim, 1998). Therefore, based on individual interview approach, the researcher ensure a comprehensive data gathering in all aspects of disaster response regarding what response activities were conduct, how and when they were operated in the aftermath of the Bam earthquake in 2003.



To ensure all aspects of disaster response are considered a questionnaire was designed based on extracted themes by reviewing
extensive literature in Persian and English languages regarding the response phase of the Bam earthquake (Figure 3). Then, semi-
structured interviews were conducted with disaster managers and experts from different governmental and non-governmental
sections in different hierarchical levels. They were selected based on the snowball sampling or chain-referral sampling method
(Biernacki and Waldorf, 1981). In This method after observing the initial subject, the researcher asks for assistance from the subject
to help identify people with a similar trait of interest. The researcher then observes the nominated subjects and continues in the
same way until a sufficient number of subjects are obtained.

---

Disaster Response Timeliness Questionnaire

1. Basic information: Sex: .............. Age:................ Education Level:...............................

2. What organization you were working in at the time of earthquake? What was your duty in the organization? Organization:...................................... Job/ Occupation:...................................

3. What was your responsibility in responding to the Bam earthquake?

4. When did you make aware of the earthquake occurrence? What did you do?

5. When first disaster management committee of your organization was held?

6. When did your organization send help to the damaged area?

7. When did you (your colleagues) reach the damaged area?

8. Alerting and warning theme:

- When and how did you find out the exact location that earthquake happened?

9. Situational awareness (damage, loss and needs assessment) theme:

- When did you get the first situational awareness report on the damaged area?

- When did you find out regarding the extent of damage and losses caused by the earthquake?

- When did you get the first report on required resources?

10. Conducting USAR operations theme:

- When and how were first USAR teams dispatched toward the damaged area and get there?

11. Evacuating casualties and providing medical services theme:

- When were injured people evacuated from the damaged area? How this happened?

- When were medical services and supplies got to the damaged area?

12. Delivering supplies and distributing resources such as food and shelters (tents) theme:

- When and how were required resources (food, water and tents) got to the damaged areas?

- When and how were resources distributed among people?

13. Burying corpses theme:

- When was burying corpses started? How was it done?





| 14. International support theme: |
| --- |
| - When was the first international help alert issued? |
| - When and how were first international SAR teams arrived at the damaged area? |
| Explanation by the researcher:……………………………………………………………… |

Figure 3. Disaster response questionnaire

**2.4. Participant recruitment**

Disaster responders (managers and experts) in main organizations who were involved in the response activities to the Bam
earthquake, and they were experienced in a variety of response activities to the other earthquake with varying degrees of education
and age were selected by the researcher using snow ball sampling method. Kerman Disaster Management Centre (KDMC) invited
the selected managers or experts for an interview. It was intended that at least thirty interviews would be necessary because of the
variety of response activities to the Bam earthquake. A tailored form of information was given to the involved interviewee outlining
the following: study objectives; the use of data; the protection of participant privacy; and the results dissemination. Individual
interviews were conducted between August to December 2011. As noted by Brod et al. (Brod et al., 2009) data collection should
continue until no new insights are produced (i.e. data saturation). In total, thirty individuals participated in the interviews to ensure
that all aspects of the response activities were covered. All interviewees were highly experienced and had been involved in
organising responses for different earthquakes, such as Zarand (-Mansourian), Bam (2003), Sirch (1981) and Golbaft (1981). Eight
of them had participated in responding to three earthquakes, and all had more than four years of experience in disaster management
and search and rescue operations, with a broad range of experiences covering many aspects of disaster-response. The average age
of the sampled individuals was 46 years, and with 83% having a university degree. A breakdown of the interviews in terms of
participants' organization, participant numbers, and their experience of earthquake response activities are documented in Table 1.

Table 1. Demographic characteristics of the study participants in the interviews

| Participants organizations | No of Participants | Average Age | level of Education | Employment status | Participants experiences |
| --- | --- | --- | --- | --- | --- |
| Paramedics and Health Services of Kerman and Bam cities | 5 | 49 | University Level | Full Time | In Zarand (2005) [48][48][48][48][45][45][45][45](45)(45)(45)(45), Bam (2003) earthquakes |





| | | | | | |
|---|---|---|---|---|---|
| Iranian Red Crescent Society of Kerman and Bam cities | 5 | 43 | University Level | Full Time | In Zarand [48][48][48][48][45][45][45] [45](45)(45)(45)(45), Bam (2003), Sirch (1981) and Golbaft (1981) earthquakes |
| Local and provincial government authorities of Kerman and Bam cities | 8 | 44 | One Diploma and Seven University Level | Full Time and Two Part Time | |
| Municipalities of Kerman and Bam cities | 5 | 49 | Two Diploma and Three University Level | Full Time | |
| Fire-fighting organizations of Kerman | 7 | 45 | Two Diploma and Five University Level | Full Time | In Zarand (2005) [48][48][48][48][45][45][45] [45](45)(45)(45)(45), Bam (2003) earthquakes |

## 2.5. Data collection

A review of existing literature on earthquake disaster response and disaster response metrics helped inform the development of a topic guide for the purpose of data collection. The topic guide covered a number of thematic areas, which sought to elicit the perceptions of participants in relation to: earthquake risk; previous earthquake experiences; levels of preparedness to response; disaster response activities and disaster response timeliness: time when responders reached the damaged area. In data collection process, the interview was conducted in a meeting room around a table facilitating face-to-face interaction with the participant. Interviews took between 30 to 60 minutes, and they were recorded digitally with the verbal consent of participants.

Kitzinger (1995) stated that participants might express a response bias, sharing opinions and viewpoints in favour of their organizations. In order to address such challenge, the interviewer tried to seek for verification and validation from participants throughout the data collection phase by asking them the same questions and repeating viewpoints back to the other relevant participants to see if answers of all interviewees in the same theme are similar. This process would assist researcher to make sure regarding "interpretive validity and research rigor" (Morse et al., 2002).



### 2.6. Analysis

Transcribing the recording by the researcher helped an initial familiarisation with the data (Braun and Clarke, 2006). Through transcribing process, all identifiable information were removed from the transcripts and the researcher have called them interview 1 (Int1) to interview 30 (Int30). The unit of analysis formed by individual lines of transcript, and each individual spoken passage contains a section in the transcripts. The coding was done manually using Excel software to be able to conduct the study using well-known Thematic Analysis (TA) method (Boyatzis, 1998; Braun and Clarke, 2006). Thematic analysis is a method for identifying, analysing, and reporting main themes in data that consists of following stages, as seen in Figure 4.

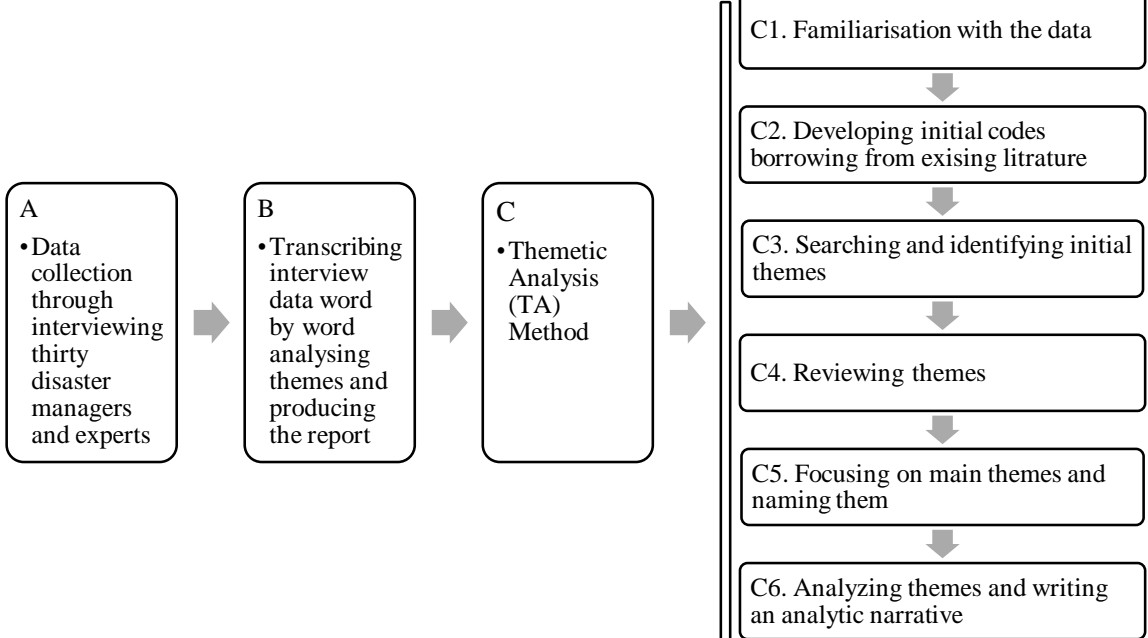

Figure 4. Thematic analysis flowchart

In order to facilitate analysing data deductively, in the early steps of analysis, "borrowing" concepts and codes from existing literature was utilized (Benaquisto and Given, 2008). Thus, the initial codes and concepts were identified by a review of existing literature documenting disaster response activities, timeliness and effectiveness in the aftermath of earthquakes, concepts associated with disaster response metrics (Abir et al., 2017). These were mainly useful in terms of organizing the data relevant to borrowed themes (e.g. 'response activities', 'warning and alerting', 'response timeliness', 'response effectiveness', 'resources distribution' etc.), specifically 'timeliness and efficacy' (Abir et al., 2017). A number of criteria were determined in order to assign data to different themes including which response activity participants was referred to, the time of reaching to the damaged area by responders of that specific response activity, describing how that activity was done in the damaged area. Thus, participant must referred to a specific response activity in their comments in order to be assigned to the specific code such as "alerting and warning,





situational awareness (damage, loss and needs assessment), conducting USAR[1] operations, evacuating casualties and providing
medical services, delivering supplies and distributing resources, burying corpses, and international support". In certain
circumstances, it was difficult to clearly assign some comments solely to one particular component of cognitive process (theme).
In such cases, the comments were attributed to relevant themes for completeness of data and then were analysed within the broader
context of the results.

## 3. Findings

By reviewing existing literature and analysing interview data, eight main themes were developed regarding disaster response timeliness in each activity in the aftermath of the Bam earthquake. These themes are described and contextualized in the following sections:

### 3.1. Timeliness in alerting and warning

Alerting and warning is all about making aware disaster responders exactly what happened and where it happened. They should be prepared always to act accordingly as fast as possible. The first alert on the epicenter of the earthquake was wrong that made responders confused on the place of being hit by the earthquake, as Jazmoorian was not populated, and there was not any dense urban areas. This may make them to think of not being quick in first hours after the earthquake. Although, disaster responders reached the Bam city around 2 hours after the earthquake:

> *"... We start riding toward Jazmoorian around 6:30 am in the morning, but on the way we saw several cars was moving toward Kerman city and one of drivers pointed at us with bloody hands and we got the point that the damage level was heavy. We reported back to Kerman center and asking for more help. We traveled around 150 km and later in the security point; they told us the earthquake center was Bam city. We reached the Bam city around 7:50 a.m...." (Int8)*

The Bam earthquake happened early morning on Friday, a weekend in Iran, therefore, many of disaster managers were not at work, they were at their home. As many interviewees report on this situation:

> *"... I was at home, and then I have woke up by the shaking, then 5 minutes after the earthquake my colleagues reported that the earthquake epicenter was in Jazmoorian. But in the meantime another colleague who was from Bam range me and said that I have tried many times to contact my family in Bam city but it seems telephone lines are busy and no one answered the phone...." (Int1)*

---

[1] Urban Search and Rescue





Few evidence shows that the earthquake epicenter was not Jazmoorian and they could see cars coming toward Kerman city that carries injuries and also people in south part of Kerman Provinces could answer their phones, but no one in Bam could answer their ones:

> *"... My relatives were in Bam city and I tried to contact them, but no one answered my calls, still I did not know where the earthquake epicenter was. We dispatched three ambulances toward south part of Kerman province with the hope of finding the exact place. Around 80 kilometer far from Kerman city, we witnessed damaged cars moving toward Kerman city asking one of them, then we found out the earthquake epicenter was in the Bam city. It was around 6:30 am .... We got to Bam around 7:30 a.m.... "* (Int4)

Making aware the involved organizations regarding the real situation in the affected area should be done quickly. It was one and half-hour later that first disaster management meeting was held in Kerman city:

> *"... The first Disaster Management meeting was held at 7:00 a.m. in Kerman city.... having all involved organizations in the meeting, here we found out the damage level in Bam city, then asked for help from other cities, it was 7:30 a.m...."* (Int5)

### 3.2. Timeliness in situational awareness (damage, loss and needs assessment)

Situational awareness means to be aware of the extent of damage and loss, and also what and how much required resources is needed to cope with the disaster. As outlined by some of the participants, it was a slow process of assessing the damaged area and reporting to top disaster managers on the real situation in Bam city. First reports were ambiguous that reported light destructions of buildings in the area that seems there was lack of conducting proper field observation before this report:

> *" ... I got to my office around 6 a.m., the first official call was from police brigade in Bam city at 6:30 reporting on the situation in Bam city as light damage level in the area. There was no news about the governor of Bam at that time..."* (Int6)

Being able to report on the situation in the affected area needs dedicated personnel who in Bam city, they were injured or killed or their relative were effected severely by the earthquake:

> *"...We could not find anyone, no mayor, no governor; everybody was missing...there were no facilities on the scene to assess the extent of damage... Many organizational personnel were killed or injured...no one was at work at 5:27 a.m. on Friday morning at the weekend..."* (Int20)

In first disaster management meeting in Kerman city, still there was not clear information on what happened in Bam. We had real observations around 8:30 a.m, and the second report on the situation of Bam city at 11:00 a.m. declaring massive destruction to buildings that caused many causalities:



*"...We flied to Bam city by a helicopter. Above Bam, we witnessed a destroyed city. It was around 9:30 a.m...."* *(Int9)*

*"...The second report on the situation in Bam city announced at 11:00 a.m. reporting extensive building damage and population loss..."* *(Int14)*

### 3.3. Timeliness in conducting Search and Rescue (SAR) operations

Conducting SAR is the most important activity in disaster response operation that can save many lives if it starts quickly. People are the first ones on scene, then they can save many others who need help, either tapped people under debris or got injured, as mentioned by interviewees:

*"... In the first hours, people saved themselves. There was no management system in place at the beginning... "* *(Int13)*

*"...We got to the scene around 11:00 a.m., but by that time many people were saved by their relatives or their neighbours..."* *(Int23)*

Professional SAR teams got to the damaged area by delay, lack of professional rescuers was a big challenge in first day of the earthquake:

*"...Many people were trapped under debris. The speed of SAR operation was slow and rescuer got to the damage area very late. Required equipment was not available. Entrapped people were dragged out of debris by force, and because of this action their spinal cord was cut that caused many people to be paralyzed (crash syndrome) forever..."* *(Int3)*

*"...SAR teams of Iranian IFRC and their rescue dogs came to Bam city on Saturday noon. They could find 670 people who were buried under debris... "* *(Int12)*

### 3.4. Timeliness in evacuating casualties and providing medical services

Evacuating injuries and providing medical services should be done quickly to ensure saving more lives. Ordinary people assisted trapped and injured people, but their help in some cases was not done in an appropriate way, as they sometimes dragged injuries out of debris by force and they carried them on a blanket or a soft bed:

*"...Many people were injured. Ordinary people carried some of injuries on a blanket to cars or ambulances and then got to the first medical centre in Bam... Sometimes injuries were dragged out of debris by force that could cause some of them crash syndrome (their spinal cord could be cut ... these people could be paralyzed for the rest of their life... (Int27)*

Many cars rushing toward Bam city that caused traffic jam and road blockage in the entrance of the city, then no one could get out of city. In this situation, air transportation could be the only option:



*"…Around 4 p.m. main road toward Bam city was blocked by many cars wanted to get into the city. No one could get out, therefore, many injuries were stuck in the traffic jams. This situation made rescue operation very slow. It was around 17 hours of road blockage at the Bam entrance road…" (Int6)*

*"…We asked people to bring their injuries to the Bam airport around 4 p.m. … Air transportation was the only way to evacuate injuries from Bam and transport them to other hospitals. It started around 10 p.m., by this time many injuries were died in the airport saloon…." (Int17)*

Being aware of all affected locations should be the aim of comprehensive disaster response plan. Involving every single place in the plan and not forgetting anywhere:

*"…We focused only on Bam and forgot other villages around the city…"(Int8)*

### 3.5. Timeliness in delivering supplies and distributing resources

Preparing supplies and resource distribution is not a primary issue in the first hours, but disaster managers should pay attention to what are the needs of people in the affected area. Enough amount of resources such as food, shelter, clothing, women needs are very important to be delivered to the area and to be distributed in an organized, and respectable way to people in need. As, it was a challenge in first days in Bam:

*"… In first day of the earthquake on Friday evening, resource distribution was random and unorganized…" (Int3)*

*"… The first supply container got to Bam around 3 p.m. including food and shelter, but it was not enough at all…" (Int17)*

Knowing where to go and how to get to people in need is a challenge. In Bam city, there were not proper street networks everywhere to service all affected people:

*"… Main streets remained running, but some of narrow alleys were completely blocked, especially in old part of the city. This caused resource distribution to be hampered. Affected people who were living near main streets could get food and tent quickly, but people who were in those narrow alleys couldn't get any resources in two first days…" (Int14)*

*"…There were many gardens in Bam city and their walls were destroyed and they were blocked the road…these blockages were hampered accessibility to far areas from main streets. People in these areas could not get enough food and shelters in first days…" (Int4)*

### 3.6. Timeliness in burying corpses



Massive destruction can kill many people in the affected area. Existence of many corpses on the affected area can endanger
people's health. Therefore, one of the disaster response action should be dealing with this situation quickly.  The Bam earthquake
caused many fatalities. Many of corpses were buried by their relatives in the Bam cemetery. In the first hours, there was no
equipment to excavate the ground and people only used shovel to grave their loves one:
*"...Many people were died under debris, and they were dragged out of debris and then buried by their relatives in the*
*Bam cemetery... In the first hours, corpses were buried in holes that excavated by hands disorderly everywhere...they*
*were not deep enough so that animals could bring them out... around 3000 corpses were buried in this condition..."(Int7)*
*"...Saturday morning around 8:00 a.m., we started to excavate canals width of 2 meters and length of 40 meters and*
*depth of 2 meters by mechanical excavators. In each canal around 140 to 150 corpses were lined up beside each other*
*and then their locations were marked by their relatives and then were covered by soil..." (Int24)*
*"... Burring corpses was started from Friday evening and continued till 5 days after the earthquake. Many people were*
*buried on the second and third days of the earthquake..." (Int24)*

**3.7.  Timeliness in international support**
Fast damage and loss assessment is key to determining the level of needed help and asking for international support. When the
extent of disaster and its damage is huge and the national government cannot handle the situation, international help alert will be
issued. In case of Bam, it happened ten hours later:
*"... First alert on international support was issued around 3:30 p.m..." (int7)*

SAR operations are the most influential part of any response plan. As, international SAR teams are equipped with advanced
technologies that can facilitate determining the location of trapped people under debris and then save more lives:
*"... SAR teams from Switzerland reached the Bam a day after the earthquake on Saturday morning at 8:00 a.m., and then*
*we had to other SAR teams on Saturday evening....there were 15 field hospitals of foreign countries that were established*
*7 days after the quake..." (Int8)*
*"... International teams were completely equipped, but our volunteers did not have even a shovel..." (Int9)*

International help such as food and clothing should be compatible with the culture of affected people:
*"... International helps (food and clothing) were not compatible with the needs of people in the Bam city" (Int9)*





Based on in-depth analysis of interview data with disaster manager and reviewing the literature as interpreted above response
activities according to the arrival time at the affected area are listed and described in Table 2 and Figure 5.
Table 2.Response time in 26 hours after the Bam earthquake

| Time | Hours after the earthquake | Explanation |
|---|---|---|
| **05:26:52 AM  (Local time)** | **0** | **Event: Earthquake alert:**<br><br>Friday, December 26, 2003 at 01:56:52 (Lam et al.) Magnitude: 6.6 Ms |
| **5:59 AM** | **35 min** | **Informing the correct location of the earthquake** |
| **5:59 AM** | **35 min** | **Provincial awareness:**<br><br> The first report was sent 35 minutes after the earthquake by the commander of the 1st Brigade of Bam |
| **7:00 AM** | **1.5h** | **Situational awareness:**<br><br> Reports on describing the situation and requesting help were sent by the commander of the 1st Brigade of Bam city |
| **7:00 AM** | **1.5h** | **Provincial disaster management meeting in Kerman city** |
| **7:00 AM** | **1.5h** | **Dispatching two ambulances from Kerman:** They were in Bam city |
| **7:30 AM** | **2h** | **Military equipment and facilities:**<br><br>Military forces dispatched to Bam city; they were in the damaged areas by **8:00 AM** (Arrival of the first team in the damaged area) |
| **8:00 AM** | **2.5h** | **Situational awareness:**<br><br>Complementary reports on describing the situation and requesting help were sent by the commander of the 1st Brigade of Bam city |





| 11:00 AM | 5.5h | **Military equipment and facilities**:<br><br>Transport of 150 medical personnel to the damaged area |
|---|---|---|
| 15:00 PM | 9.5h | **Evacuation of casualties by airplanes:**<br><br>The evacuation was done via Bam airport. |
| 15:30 PM | 10h | **International alert:**<br><br>The Iranian authorities launched a request for international assistance. |
| 16:00 PM | 10.5h | **Military equipment and facilities:**<br><br>On the day of the disaster, 937 medical assistance personnel were transferred to the damaged area.<br><br>The first Iranian field hospital was set up by Iranian military forces in the area on the first day. |
| 17:00 PM | 11.5h | **Arrival of ambulances and surgical teams:**<br><br>102 ambulances, 8 field emergency stations, and 4 surgical teams arrived in the damaged area. |
| 19:00 PM | 13h | **Arrival of specialized medical care teams:**<br><br>They arrived in the city almost 14 hours after the earthquake. |
| 12:00 PM (Midnight) | 18.5h | **Arrival of ambulances:**<br><br>24 self-sufficient ambulances and vans arrived in the damaged area. |
| 7:00 AM on 27 December 2003 | 25.5h | **Arrival of international Urban Search and Rescue (USAR) team:**<br><br>Switzerland provided the first Urban Search and Rescue team, who arrived at Bam airport on the morning of 27 December.<br><br>USAR teams from 10 countries arrived in Bam on 27 December late evening. |



Figure 5.The earthquake disaster-response time line (in 26 hours after the earthquake)





## 4. Discussions

According to interviews data, only main themes regarding response activities, during the 26 hours after the Bam earthquake are described. These include: alerting and warning, situational awareness (damage, loss and needs assessment), conducting USAR operations, evacuating casualties and providing medical services, delivering supplies and distributing resources such as food and shelters (tents), and burying corpses, and, finally, international support.

### 4.1. Alerting and warning

The first alert was issued by the Geophysical Institute of Tehran University (Geophysical Institute of Tehran University (Gitu), 2003), reporting an earthquake occurrence in south of Kerman province in Jazmoorian region, an area about 200 km from the real earthquake's location. However, about 35 minutes later, according to the Disaster Management Task Forces of Kerman province, and reports coming from the military commander of Bam city, the earthquake epicenter was precisely identified in Bam city. It caused a delay in informing the involved organizations in dispatching their resources to the damaged areas. Additionally, the delay in responding to the earthquake happened because of the lack of proper information on the extent of damage to buildings and population (the situation) in the affected area (Naserasadi, 2004; Berberian, 2009) because some communication towers located on rooftops of collapsed buildings were damaged (Nadim et al., 2004; Manafpour, 2003; International Federation of Red Cross and Red Crescent Societies (Ifrc), 2004). As a result, the telephone lines, mobile phone networks, and Internet networks were disrupted after the earthquake.

### 4.2. Situational awarness (assessing damage, loss and needs)

The primary information on the impact of the earthquake on the building damage and population losses were reported by the military commander about 35 minutes after the earthquake. The initial assessments were not of much concern and disaster managers thought that damage was negligible or very light; this was because of the lack of information on the ground. However, complementary reports on the situation, the damage to buildings and infrastructures, and population losses 1 and 2 hours after the disaster, indicated the real critical situation on the ground.

The first official map of the building damage in the city was produced using aerial photography two days after the earthquake on 28[th] of December (Alavi Rzavi, 2008; Manafpour, 2003). Furthermore, the United Nations Disaster Assessment and Coordination (UNDAC) teams arrived early on the morning of 28 December and performed the first rapid assessments of the disaster zone to estimate the resources required in assisting the victims of the earthquake.

### 4.3. Conducting USAR operations

The emergency response services of the city (the governor's office, fire services, police stations, and health and medical services) were all severely affected. Many people working in these public sectors were either killed or injured. The Governor's office, which was supposed to be the disaster management centre, was severely damaged. The main firefighting centre and some



of the police stations had completely collapsed (Nadim et al., 2004; International Federation of Red Cross and Red Crescent Societies (Ifrc), 2004; Akbari et al., 2004; Manafpour, 2003). The impact of the earthquake on the population, buildings, and infrastructure was extensive. Consequently, hampered the functionality of local emergency management services and postponed the disaster-response activities in Bam city.

In the early morning hours, after the Bam earthquake, the survivors desperately tried to help their family members and neighbors who were missing or trapped under the debris of collapsed buildings. Within half an hour of the earthquake, the Iranian Red Crescent Society (IRCS) began to mobilize its emergency response teams and, within two hours, the first IRCS urban search and rescue (USAR) teams had reached Bam city. Moreover, the first assistance teams of Iranian military forces arrived to the damaged area 2 hours after the earthquake. It was followed by the arrival of military forces of different brigades of the Kerman Province in the area at 11 AM. In addition to the official operations, many volunteers from various segments of Iranian society came to the region to help those affected.

### 4.4. Evacuating casualties and providing medical services

All three main hospitals and 10 urban health centres were destroyed and rendered unusable. Many people working in these public sectors were either killed or injured. Due to the extensive damage and unavailability of local healthcare workers, only one health facility was functional in Bam city after the earthquake (International Federation of Red Cross and Red Crescent Societies (Ifrc), 2004; Earthquake Engineering Research Institute (Eeri), 2004).

The majority of the evacuated victims (serious injuries) were transferred to first-line treatment centers by ambulances and other vehicles. They have been admitted to 12 hospitals in the neighbouring cities and big cities in Iran. Many of these treatment centers were located in the cities of Kerman and Jiroft, 185 km and 120 km far from Bam city, respectively.

Many casualties were transported to the airport as the second major settlement center before air evacuation. The injuries were examined by medical professionals at the airport and the level of medical care required was identified and categorized as red (urgent), yellow (delayed or expectant), green (minor), or black (deceased) (Mohebbi et al., 2008). The air evacuation started around 3 PM, about 9.5 hours after the earthquake occurred. In the first 48 hours after the earthquake struck, more than half of the 23,000 injured people were evacuated to large hospitals around Iran using civilian and military aircrafts, helicopters (air evacuation), ambulances and private cars (road transport). According to a study by Mirhashemi et al. (2007), 41.7% of the patients were transported to hospitals by air, 15.1% by private vehicles, and 22.7% by ambulance. It took, on average, 25.2 ±16.3 hours to reach a hospital. For example, the first group of patients reached the Chamran Hospital in Shiraz just 12 hours after the earthquake (Emami et al., 2005).

In addition to twelve large hospitals in neighboring provinces, a dozen field hospitals provided by international relief medical groups and the Red Crescent Society were involved in providing medical care to the victims (Abolghasemi et al., 2005). For





example, the first Iranian field hospital was set up by Iranian military forces on the first day at 11 PM, and 937 medical assistance personnel were transferred to the area (Abolghasemi et al., 2006).

### 4.5. Delivering supplies and distributing resources

The infrastructure of Bam city suffered a range of slight to heavy damage. The distributing network of drinking water and electricity distribution system were completely damaged (Alavi Rzavi, 2008). This left people in the damaged area without water and electricity.The main streets in Bam city were still usable immediately after the earthquake; however, the debris of collapsed buildings blocked 70% of the narrow streets close to the center of the city for several days. The airport runway and railroad only suffered slight damage (Nadim et al., 2004), which meant they could be used for dispatching supplies to the city immediately after the earthquake (Alavi Rzavi, 2008).

Therefore, resource distribution, spatially providing shelter in the early days was essential for survivors of the earthquake who were suffering from extreme emotional and physical distress, due to the cold weather at night, sometimes below freezing temperatures during the winter (Fallahi, 2007). The Iran Red Crescent Society (IRCS) had provided temporary shelters in the form of tents and food. They began distributing more than 50,000 tents, as emergency shelters, and food (canned foods) among those made homeless by the earthquake as early as the first day of the earthquake (Ghafory-Ashtiany and Hosseini, 2008). Most people preferred to set up their tents next to their destroyed or damaged houses; others were accommodated in existing open areas.

In order to manage the relief efforts, the affected region including Bam and Baravat cities, and Spikan, Poshtehrood, Corek and Baghchemak villages were divided into 14 zones and all the humanitarian efforts in each zones were coordinated with authorities in one of the provinces in the country. Later, this was reconsidered, and Bam city was divided into 10 zones, including 49 sub-zones. This appeared to be an effective approach for managing relief efforts in a large disaster zone (Ghafory-Ashtiany and Hosseini, 2008; Alavi Rzavi, 2008). The relief efforts and resources distribution had happened quickly after the earthquake. However, people located beside main streets received most of the resources first, while others, who were further from the main streets, were reached after a period of several days.

### 4.6. Burying corpses

The main causes of morbidity and mortality were direct injuries, fractures, traumatic injuries, and suffocation as a result of building collapses (Abolghasemi et al., 2006). The deceased people were buried through traditional religious ceremonies in collective graves (Ghafory-Ashtiany and Hosseini, 2008); about 93% of corpses were buried within 48 hours after the earthquake (Akbari et al., 2004).

### 4.7. International support

The Iranian authorities did not make the first request for international assistance until 03:30PM (Abolghasemi et al., 2006) about 10 hours after the earthquake. The first international notification of the earthquake was posted on the Virtual On-Site Operations Coordination Centre (Virtual OSOCC) at 03:42 AM Universal Time Coordinate (Lam et al.).



Within 72 hours of the earthquake, the UN dispatched its Disaster Assessment and Coordination Team (UNDAC) to support the Iranian Government in coordinating this enormous international response. The UN Country Team and UN agencies provided relief items, as well as technical support. The International Federation of Red Cross and Red Crescent Societies (IFRC) and various Non-Governmental Organizations (NGOs) set up field hospitals and distributed food items, blankets, and tents as emergency shelters (United Nation (Un), 2004; Fallahi, 2007). In response to the earthquake, international agencies also were mobilized from many countries and they contributed to the search and rescue operations. Nearly 40 international teams provided USAR operations in Bam, but only five of them arrived within 30 hours after the earthquake. The response and cooperation between Iranian authorities, Iranian Red Crescent Society (IRCS) and the international community was exemplary (United States Agency for International Development (Usaid), 2004). However, the military forces, operational teams and volunteered people did not know where to go first in the damaged area, and they were distributed randomly to different areas.

## 5. Conclusions

By interviewing disaster managers and reviewing the literature the timeliness of main response activities to the Bam earthquake was investigated in 26 hours after the earthquake occurrence aiming to understand when each activity was conducted and what was the reasons of having delays, as explained in the following:

-Response activities should be conducted quickly and accurately to facilitates assisting people in need. Disaster responders must be prepared always to act accordingly in case of any disaster occurrence.

-The results showed in responding to the Bam earthquake, disaster managers were not prepared for such a devastating event, and more importantly, the country's disaster management system was not prepared to cope with a disaster of this scale. The extensive damage to buildings and infrastructure, accompanied by the large number of casualties, challenged the local emergency response services. The problem was compounded by the extensive damage to the utilities (emergency services) and lack of personnel (people who have been working in different sectors in Bam city such as doctors, nurses, rescuers etc.). These caused delays in reporting on the situation of damage and losses in first place.

-There was not any disaster management center in Bam city in first day in order to manage disaster response operations or event to store the resources first or assign SAR teams to the highly damaged areas. SAR teams, vehicles carrying resources and volunteers were distributed randomly to the damaged areas in Bam city. Lack of an operation center in the area caused delays in conducting all activities in disaster response.

-Defining the level of needed help was a critical act. The first request for international assistance was launched about 10 hours after the earthquake. This delay in issuing the request was due to the lack of information on damage level and required resources,



which significantly affected response time. This showed that the situational awareness was reported without conducting any proper field observations in the first hours.

-People were the first on the scene to help. However, people who were not prepared to cope with the earthquake. Although, they were helping the others immediately, but they had not enough training on first aid learning, rescuing victims and carrying them. Their response timeliness was quick and extraordinary that had significant effect in saving many lives.

-Security issues was another challenge, as there were many traffic jams in the main road toward Bam city that hampered response operation via roads. This needs more investigations to shed light on what was the reason of having this situation in first day of the earthquake

-Developing a response strategic plan prior to earthquakes occurrence is an efficient way of dealing with a disaster. Therefore, the results of this study that considered the real situation and deficit in responding to a disaster in Iran; can assist disaster manager in designing a proper action plan (strategic response plan) in responding to a similar earthquake in the region incorporating the Sendai Framework for Disaster Risk Reduction.

**Acknowledgment**

The author very much appreciate the support by the Graduate University of Advanced Technology and Disaster Management Cneter of Kerman Manucipality.

**Competing interests**: The authors declare that they have no conflict of interest.

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
