# Peer review of "Earthquake Response Timeliness: Disaster Managers Experience in Responding to Earthquakes in Iran"

_Natural Hazards and Earth System Sciences, 2023_

## Author Comment (AC1)

Dear Editor,

I greatly appreciate the valuable comments of reviewer. I have tried to incorporate changes to reflect all of the suggestions provided by the reviewers.

I have highlighted the changes within the manuscript. Here is a point-by-point response to the reviewers' comments.

We hope the manuscript after careful revisions meet your high standards.

Yours sincerely,

RC1: (C: Reviever's comment)
C: This manuscript evaluates societal response timeliness on earthquake disasters, using the 2003 Bam earthquake in Iran as an example. The data used for the evaluation consist of interviews with several actors in the emergency response chain that is processed and structured to extract significant information on the timeliness of different response actions. The paper concludes with general and specific recommendations on earthquake disaster response measures. Although the paper is readable, it requires extensive spell-checking before resubmission. The Figures are partly of inacceptable quality.

Answer: I have read the paper several times and revised all spelling mistakes. Also, I have increased the quality of the figures and replaced them in the text.

C: The scientific work presented belongs to the field of social sciences and does not exploit geoscientific or at least environmental information. It is solely based on interview information and uses socio-scientific methods, where I am not sure the general audience of NHESS is familiar with. Consequently, I may suggest to present this work to a journal in the field of social sciences. If, however, the work would be suitable for NHESS, I would recommend the authors to present the data and especially the methodology applied in much more detail, also accompanied by appropriate illustrations and presentations of examples.

Answer: I have tried to review response phase activities timeliness in the aftermath of the Bam earthquaks. This paper will give readers an idea on what are the main respose activities and how they should be done in a timely manner. It will open up new questions

regarding response activities by evelauting real situation in the Bam City after the earthquake occurrence.

C: Abstract: Please specify Magnitude, Location and Date of the Bam earthquake when first mentioning it.

Answer: I have revised the abstract as shown in the following:

Abstract. This paper explores the timeliness of main response activities in the Bam earthquake in order to shed light in disaster response effectiveness in the aftermath of the earthquake. Disaster managers who were involved in disaster response activities in the Bam earthquake (on 26 December 2003 with the magnitude of 6.3 mb) were selected using snowball sampling method and 30 participants were interviewed.

C: P107: The mentioned "Thematic analysis" is not well-know to a geoscientific community, I may suppose. Please explain.

Answer: I have explained "Thematic analysis" in line number 194 - 206  as mentioned in the following:

Transcribing the recording by the researcher helped an initial familiarisation with the data (Braun and Clarke, 2006). Through the transcribing process, all identifiable information was removed from the transcripts and the researcher called them interview 1 (Int1) to interview 30 (Int30). The unit of analysis is formed by individual lines of the transcript, and eachspoken passage contains a section in the transcripts. The coding was done manually using Excel software to be able to conduct the study using the well-known Thematic Analysis (TA) method (Boyatzis, 1998; Braun and Clarke, 2006). Thematic analysis is a method for identifying, analyzing, and reporting main themes in data that consists of the following stages, as seen in Figure 4. Based on this method, qualitative data were analized with aim of looking for patterns in the meaning of the data to find themes. "It is an active process of reflexivity in which the researcher's subjective experience is at the center of making sense of the data" (Braun and Clarke, 2006).

[Figure]

Figure 4. Thematic analysis flowchart

In order to facilitate analysing data deductively, in the early steps of analysis, "borrowing" concepts and codes from existing literature was utilized (Benaquisto and Given, 2008). Thus, the initial codes and concepts were identified by a review of existing literature documenting disaster response activities, timeliness and effectiveness in the aftermath of earthquakes, and concepts associated with disaster response metrics (Abir et al., 2017). These were mainly useful in terms of organizing the data relevant to borrowed themes (e.g. 'response activities', 'warning and alerting', 'response timeliness', 'response effectiveness', 'resources distribution' , etc.), specifically 'timeliness and efficacy' (Abir et al., 2017).

C: P153: Please explain "snowball sampling" applied to your data sufficiently using an example.

Answer: I have explained "snowball sampling" in line  153   and line 163  as follow:

They were selected based on the snowball sampling or chain-referral sampling method (Biernacki and Waldorf, 1981). In this method after observing the initial subject, the researcher asks for assistance from the subject to help identify people with a similar trait of interest. The researcher then observes the nominated subjects and continues in the same way until a sufficient number of subjects are obtained.

Disaster responders (managers and experts) in main organizations who were involved in the response activities to the Bam earthquake, and they were experienced in a

variety of response activities to the other earthquake with varying degrees of education and age were selected by the researcher using snowball sampling method. This is a non-probability sampling method where new interviewee are introduced by other interviewees to form part of the sample.

C: P207: Which "criteria"? How many? Please specify.

Answer: This section was revised in line 213.

Three criteria were determined in order to assign data to different themes. These criteria were included as mentioned in the following: 1. which response activity participants was referred to, 2. the time of reaching to the damaged area by responders of that specific response activity, and 3. describing how that activity was done in the damaged area.

C: P213: How was the assignment done? Automatically? Manually? Please explain and provide examples.

Answer: This was mentioned in line 198.

The coding was done manually using Excel software to be able to conduct the study using well-known Thematic Analysis (TA) method (Boyatzis, 1998; Braun and Clarke, 2006).

C: P216: Please supply a list or a Figure showing "thematic analysis": "themes", "criteria", "assignement"…

Answer: This was mentioned in line 215.

Thus, participant must referred to a specific response activity in their comments in order to be assigned to the specific code such as "alerting and warning, situational awareness (damage, loss and needs assessment), conducting USAR operations, evacuating casualties and providing medical services, delivering supplies and distributing resources, burying corpses, and international support". In certain circumstances, it was difficult to clearly assign some comments solely to one particular component of cognitive process (theme). In such cases, the comments were attributed to relevant themes for completeness of data and then were analysed within the broader context of the results.

[Figure]

Figure 4. Thematic analysis flowchart

C: Figure 2: What is shown in the map?

Answer: I have redesigned the map. It is shown the geographical location of Bam city in Kerman Province, Southwest of Iran.

[Figure]

Figure 2. The geographical location of Bam city in Kerman Province, Southeast of Iran.